# Study on the Inhibitory Effect of Bioactive Peptides Derived from Yak Milk Cheese on Cholesterol Esterase

**DOI:** 10.3390/foods13182970

**Published:** 2024-09-19

**Authors:** Peng Wang, Xuemei Song, Qi Liang

**Affiliations:** Functional Dairy Products Engineering Laboratory of Gansu Province, College of Food Science and Engineering, Gansu Agricultural University, Lanzhou 730070, China; wangpeng3335@163.com (P.W.); springwinter110@126.com (X.S.)

**Keywords:** yak milk cheese, peptide, molecular docking, molecular dynamics simulation, cholesterol esterase inhibitory activity

## Abstract

The bioactive peptides derived from yak milk cheese exhibited cholesterol-lowering properties. However, there was limited research on their inhibitory effects on cholesterol esterase (CE) and elucidation of their potential inhibitory mechanisms. In this study, we identified CE-inhibiting peptides through virtual screening and in vitro assays. Additionally, molecular docking and molecular dynamics studies were conducted to explore the mechanisms. The results indicated that peptides RK7 (RPKHPIK), KQ7 (KVLPVPQ), QP13 (QEPVLGPVRGPFP), TL9 (TPVVVPPFL), VN10 (VYPFPGPIPN), LQ10 (LPPTVMFPPQ), and SN12 (SLVYPFPGPIPN) possessed molecular weights of less than 1.5 kDa and a high proportion of hydrophobic amino acids, demonstrating notable inhibitory effects on CE. Molecular docking and dynamics revealed that peptides RK7, KQ7, QP13, and VN10 bound to key amino acid residues Arg423, His435, and Ser422 of CE through hydrogen bonds, hydrophobic interactions, salt bridges, and π–π stacking, occupying the substrate-binding site and exerting inhibitory effects on CE. The four peptides were further synthesized to verify their CE-inhibitory effects in vitro. RK7, KQ7, QP13, and VN10 exhibited inhibitory activity on CE with IC_50_ values of 8.16 × 10^−7^ mol/L, 8.10 × 10^−7^ mol/L, 4.63 × 10^−7^ mol/L, and 7.97 × 10^−7^ mol/L; RK7, KQ7, QP13, and VN10 were effective in inhibiting CE after simulated gastrointestinal digestion, especially with a significant increase in the inhibitory activity of KQ7 and RK7, respectively. Our findings suggested that bioactive peptides from yak milk cheese represented a novel class of potential CE inhibitors.

## 1. Introduction

Cardiovascular disease (CVD) has represented a serious threat to human health in recent years. In Europe, cardiovascular disease accounted for 45% (2.2 million per year) of mortality in women, while it accounted for 39% (1.9 million per year) of mortality in men. CVD ranked first in disease mortality among urban and rural residents in China, accounting for 48.00% and 45.86% of rural and urban deaths, respectively [1]. It was well known that an increase in serum cholesterol levels could trigger diseases such as heart disease, atherosclerosis, obesity, and diabetes [2]. Intake of cholesterol from the diet significantly affected serum cholesterol levels [3]. Cholesterol esterase was an important serine hydrolase that played a key role in the absorption of dietary cholesterol and its transport from cholesterol micelles to intestinal cells. Due to its important role in absorption and transport, inhibition of cholesterol esterase was a potential approach for treating hypercholesterolemia and atherosclerosis. Studies such as that by Allen [4] had shown that compounds that effectively inhibited cholesterol esterase were sometimes accompanied by serious side effects and, thus, the study of food-borne cholesterol esterase inhibitors had attracted a great deal of attention.

In addition to being rich in nutrients, bovine milk protein also contained peptide sequences capable of regulating specific biological activities, known as bioactive peptides [5]. During the ripening of yak milk cheese, the milk protein was subjected to the action of peptidase and a fermentation agent, which produced small molecule, milk-derived peptides with various bioactive functions [6]. Milk-derived bioactive peptides had potential bioactivity in functional and nutritional food formulations and exerted beneficial effects such as mineral binding, immunomodulation, anti-thrombosis, and anti-hypertensive functions, among others [7]. Previous studies by the group had shown that the primary structures of the peptides produced by the degradation of yak milk cheese casein in this experiment played an important role in peptide bioactivity. These peptides were also shown to have a number of beneficial bioactive functions, such as antiglycan activity [8], bacteriostasis [9], antioxidant [10], and Angiotensin-Converting Enzyme (ACE) activity inhibition [11]. However, the peptides were not found to inhibit the activity of cholesterol esterase or have any inhibitory mechanisms against the action of cholesterol esterase.

In recent years, virtual screening has been successfully applied during the high-throughput screening of enzyme inhibitors to predict interactions between inhibitors and proteins [12]. This has been combined with accurate and efficient studies of peptide bioactivity on bioinformatics platforms such as the ChemDraw, Peptide DB, and BIOPEP–UWM databases. Discovery Studio can be used to quickly and efficiently predict the initial docking pose between the peptide and cholesterol esterase, and then molecular dynamics simulations can be performed using GROMACS 2022.3 software [13] to study and improve the structural stability of the complexes, kinetic properties, and other factors. These results can be used to jointly elucidate the mechanism of action of the bioactive peptide through the analysis of data from the molecular dynamics simulation [14].

The present study focused on peptides RK7, KQ7, QP13, TL9, VN10, SN12, and LQ10 that were extracted from yak milk cheese using the chloroform–methanol method, and isolated and purified using Sephadex G-25 dextran gel chromatography [15]. The molecular dynamics analysis elucidated the mechanism of action of cholesterol esterase-inhibiting peptides through bioinformatics analysis of the physicochemical properties of the above peptides combined with molecular docking. In addition, in vitro activity experiments were conducted to provide theoretical data to study the inhibition rate of cholesterol esterification inhibitory peptides.

## 2. Materials and Methods

### 2.1. Materials and Reagents

The peptides RK7, KQ7, QP13, TL9, VN10, SN12, and LQ10 all come from mature yak milk cheese.

Synthetic peptides RK7, KQ7, VN10, and QP13 which were required for the in vitro validation assays were provided by Sangon Biotech Co., Ltd. (Shanghai, China).

Porcine cholesterol esterase and the ELISA kit were provided by Jiangsu JINGMEI Biotechnology Co., Ltd. (Yancheng, China).

Atorvastatin was obtained from Qilu Pharmaceutical Co., Ltd. (Jinan, China).

### 2.2. Instruments and Equipment

This study used Discovery Studio Client v16.1.0 (DS) Accelys Inc. (San Diego, CA, USA). ChemDraw Professional 20.0 (CD) CambridgeSoft Inc. (Cambridge, MA, USA). The microplate reader (VersaMax) and vortex mixer (VM–500S) were provided by Quan’an Experimental Instrument Co., Ltd. (Ningbo, China). The pH meter (PHS–3C) was manufactured by Shanghai YiDian Scientific Instrument Co., Ltd. (Shanghai, China). The analytical balance (FR224CN) was supplied by Ohaus Instrument Co., Ltd. (Shanghai, China). The thermostatic water bath (HWS26) was obtained from Shanghai Yiheng Technology Co., Ltd. (Shanghai, China).

### 2.3. Experimental Methods

#### 2.3.1. Prediction of Physicochemical Properties of Peptides RK7, KQ7, QP13, VN10, TL9, LQ10, and SN12

The isoelectric point (pI), net charge, and theoretical relative molecular weight of the peptides were determined using the Peptide Property Calculator platform. The instability indices of the seven peptides were determined using the ExPASy ProtParam platform. The hydrophobicity and proportion of hydrophobic amino acids of the seven peptides were determined using the PEPTIDE 2.0 CE platform [16].

#### 2.3.2. Structure Optimization and Processing of Peptides and Cholesterol Esterase

The molecular structures of peptides RK7, KQ7, QP13, TL9, VN10, LQ10, and SN12 were drawn using ChemDraw Professional 20.0 software, as shown in Figure 1. The molecular structures were opened in Discovery Studio Client v16.1.0 software and generated as conformers using the Prepare Ligands function under ‘Small Molecules’. Subsequently, all compounds were energetically minimized using the CHARMM force field in preparation for molecular docking with receptors at a later stage. In this experiment, all seven peptides existed in an irregularly coiled form, and the fact that the peptide chains existed in an irregularly coiled form meant that they had a high degree of conformational flexibility in the free state. This flexibility allowed them to adjust their structure by interacting with target molecules, such as receptors, enzymes, or other proteins. Unlike the conventional alpha-helix or beta-folding, irregularly coiled peptides were more adaptable, and this flexibility might endow these peptides with additional functional potential, such as signaling, protein recognition, or receptor activation. This diversity was also common in biologically active peptides with functional specificity (e.g., antimicrobial peptides), which could interact with different biomolecules. Peptide RK7 (Arg-Pro-Lys-His-Pro-Ile-Lys) yielded two reciprocal isomers and one stereochemical isomer, while peptides KQ7 (Lys-Val-Leu-Pro-Val-Pro-Gln), QP13 (Gln-Glu-Pro-Val-Leu-Gly-Pro-Val-Arg-Gly-Pro-Phe-Pro), TL9 (Thr-Pro-Val-Val-Val-Pro-Pro-Phe-Leu), VN10 (Val-Tyr-Pro-Phe-Pro-Gly-Pro-Ile-Pro-Asn), LQ10 (Leu-Pro-Pro-Thr-Val-Met-Phe-Pro-Pro-Gln), and SN12 (Ser-Leu-Val-Tyr-Pro-Phe-Pro-Gly-Pro-Ile-Pro-Asn) yielded only one reciprocal isomer and one stereochemical isomer. This may reflect the influence of the amino acid composition of the different peptide sequences on their dynamic conformational states. The formation of the reciprocal isomers was related to the presence and arrangement of specific amino acids in the peptide such as proline (Pro) residues and other residues with a high degree of rotational capacity such as lysine (Lys) [17]. These amino acids were capable of causing rapid, local structural changes. Each peptide gave rise to one stereochemical isomer, and the presence of stereochemical isomers might have had a direct impact on the biological activity of the peptide, as different isomers might have different binding affinities or biological effects. The presence of stereochemical isomers might determine the specific recognition of these peptides with target molecules, the efficiency of their binding, and their activity in signaling or catalysis in biological systems [18].

The 3D structure of cholesterol esterase is shown in Figure 2. The crystal structure of CE with PDB ID: 1AQL was obtained from the RCSB Protein Data Bank https://www.rcsb.org (accessed on 18 March 2024). We opened the acquired protein file using Discovery Studio Client v16.1.0 software, eliminated aqueous molecules and heteroatoms from the three-dimensional configuration, refined the protein conformation utilizing the ‘Clean Protein’ function within the Macromolecules module, and hydrotreated with Hydorgens, followed by the energy minimization of all compounds under the CHARMM force field, which re-fined the protein architecture.

#### 2.3.3. Molecular Docking

RK7, KQ7, QP13, TL9, VN10, SN12, and LQ10 were sequentially LibDocked molecularly to CE (PDB: 1AQL) using the ‘Dock Ligands’ (LibDock) mode under the ‘Receptor-Ligand Interactions’ window in the Discovery Studio Professional 20.0 docking software. None of the peptide–enzyme complex models showed significant differences in their 3D distance constraints. Therefore, the 3D and 2D structures of peptide–enzyme complexes, LibDock Score docking scores, and docking active sites were generated based on successful docking with the best model, thus, determining the optimal orientation of the active sites of the peptide and CE [17]. After successful docking and the generation of a 2D graph in the ‘View Interactions’ mode, the amino acid residues in the peptide–enzyme complex and intermolecular interaction forces, such as van der Waals, hydrogen, and hydrophobic interactions, can be visually observed in Discovery Studio Professional 20.0 software.

#### 2.3.4. GROMACS Molecular Dynamics Simulation

The molecular dynamics experiment in this study was conducted using GROMACS 2022.3 software [13,18]. A GAFF force field was applied to the peptides RK7, KQ7, QP13, and VN10 using AmberTools22 for preprocessing. Hydrogenation and computation of RESP potentials for peptides RK7, KQ7, QP13, and VN10 were performed using Gaussian 16W. The potential energy data were added to the topology file of the molecular dynamics system. The simulation conditions were a static temperature of 300 K and an atmospheric pressure of 1 bar. Amber99sbildn was used as the force field, with water molecules as the solvent (TIP3P water model). The total charge in the simulation system was neutralized by adding an appropriate number of Na^+^ ions. The simulation system was minimized using the steepest descent method for energy minimization, followed by 100,000 steps of equilibration under constant temperature and volume (NVT) and constant temperature and pressure (NPT), with a coupling constant of 0.1 ns and a duration of 100 ns each. Finally, a free molecular dynamics simulation was performed, consisting of 5,000,000 steps, with a time step of 2 ns, and a total duration of 100 ns (nanoseconds). After the simulation was completed, trajectory analysis was performed using GROMACS 2022.3 software to calculate the root-mean-square deviation (RMSD), root-mean-square fluctuation (RMSF), hydrogen bonds (HBond), solvent-accessible surface area (SASA), and binding free energy (MM/GBSA) of each amino acid’s motion trajectory. The binding between the enzyme and peptides was investigated to evaluate the interaction forces.

The binding free energy (MM/GBSA) was calculated by subtracting the free energy of unbound small molecule ligands and receptor proteins from the free energy of the complex:(1)$$\Delta G_{\mathrm{MMGBSA},\mathrm{solvated}}=\Delta G_{\mathrm{complex},\mathrm{solvated}}\left[\Delta G_{\mathrm{receptor},\mathrm{solvated}}+\Delta G_{\mathrm{ligand},\mathrm{solvated}}\right]$$

Then, the free energy change was calculated using the following equation:ΔG_solvated_ = E_gas_ + ΔG_solvation_ − TS_solute_(2)
where ΔG_solvated_ represented the true free energy, and the solvent was averaged due to the use of the implicit solvent model. Gas-phase energies () E_gas_ were typically derived from molecular mechanical (MM) energies in the force field, and solvated free energies () ΔG_solvated_ were calculated using an implicit solvent model and further decomposed into the sum of electrostatic and non-polar contributions.

#### 2.3.5. Synthesis and CE Inhibitory Activity Assay of Peptides RK7, KQ7, QP13, and VN10

The peptide TL9 could not be synthetically purified and synthesized due to its fully hydrophobic nature, as indicated by the results of the above computer simulations. In this experiment, four peptides, RK7, KQ7, QP13, and VN10, were synthesized through solid-phase synthesis, and the purity and molecular weight of the peptides were analyzed using *LC–MS*. This experimental procedure was performed by Sangon Biotech Co., Ltd. (Shanghai, China).

CE inhibitory activity was determined using spectrophotometry and the CE inhibition rate was calculated according to the following formula:(3)$$\mathrm{CE}\;\mathrm{inhibition}\;\mathrm{rate}/\%=\left(1-\frac{\mathrm{ODs}-\mathrm{ODsb}}{\mathrm{ODc}-\mathrm{ODb}}\right)\times 100$$

In the formula, *ODs:* is the absorbance value of the sample group; *ODsb*: is the absorbance value of the sample blank group; *ODc*: is the absorbance value of the control group; and *ODb*: is the absorbance value of the blank group.

#### 2.3.6. Simulating the Impact of Gastrointestinal Digestion on α-Amylase Inhibitory Activity

We dissolved the synthesized peptide aqueous solution in pepsin (enzyme: substrate 1:50 *w*/*w*) and adjusted the pH to 2 using 1 mol/L HCl. Then, we hydrolyzed this at 37 °C for 90 min, then adjusted the pH to 7.5 with 1 mol/L NaOH solution and added trypsin (enzyme: substrate 1:50 *w*/*w*). Next, we hydrolyzed this at 37 °C for 180 min in a constant temperature water bath. Subsequently, we inactivated this by boiling it in a 95 °C water bath for 10 min. Finally, we took samples every 30 min and stored each sample at −20 °C for later analysis [19].

The determination of CE activity refers to Section 2.3.5.

#### 2.3.7. Data Processing

Each experimental sample was tested in triplicate, and the data were processed using IBM SPSS Statistics 26 software. Figures were generated using Origin 2018 and GraphPad Prism 8.0.2 software.

## 3. Results and Discussion

### 3.1. Physicochemical Characterisation of RK7, KQ7 QP13, TL9, VN10, SN12, and LQ10 Peptides

The molecular weight of peptides was a crucial determinant of their biological activity. Research had shown that peptides with lower molecular weights demonstrated greater efficacy in exhibiting biological activity [20,21,22,23]. As illustrated in Table 1, the molecular weights of RK7, KQ7, QP13, TL9, VN10, SN12, and LQ10 ranged from 779.50 to 1392.69 Da, all of which were relatively modest. This likely contributed to their heightened biological activity, which was consistent with the observed increased cholesterol-lowering effects of smaller molecular weight peptides in this study. Hydrophobic amino acids, such as leucine (Leu), alanine (Ala), and proline (Pro), were essential for the biological activity of peptides. The quantity of these amino acids directly affected the peptide’s binding affinity to cholesterol receptors. Moreover, it had been observed that the cholesterol-lowering activity of peptides related to their molecular weight; smaller peptides were able to exert biological activity more effectively. This underscores the significance of hydrophobic amino acids in peptide biological activity [24], aligning with the fact that peptides rich in hydrophobic amino acids exhibited higher activity in this study. Among the peptides studied, RK7, KQ7, QP13, TL9, VN10, SN12, and LQ10 all contained a substantial amount of hydrophobic amino acids, as seen in Table 1, which might account for their cholesterol-lowering effects. This study indicated that the investigated peptides contained hydrophobic amino acids such as Leu, Pro, Val, and Gly, and their quantities were closely related to their cholesterol-lowering activity. Consequently, a biological mechanism could be hypothesized wherein the binding site of the peptide to the cholesterol receptor might involve a specific interaction with hydrophobic amino acids, thereby enhancing the peptide’s cholesterol-lowering efficacy. This hypothesis necessitates further experimental validation.

### 3.2. Molecular Docking Using LibDock

#### 3.2.1. Reliability Verification of Molecular Docking Methods

The structures of 1AQL co-crystal complexes were downloaded from the PDB database, and then docked and compared with the co-crystal complexes. The RMSDs of the complexes were all <2 Å: A–TCH: 1.8141 Å. The conformations of the ligands in the original crystal structures of the target proteins overlapped with those of the docked ligands; these are shown in Figure 3. The conformations of the four docked ligands were essentially overlapped with the conformations of the ligands in the original crystal structures. The overlapping conformations and RMSD values indicate that the docking method and parameter settings of this molecule are reliable.

#### 3.2.2. Molecular Docking of CE Inhibitory Peptides

Molecular docking was employed to simulate the binding interactions responsible for the association between CE and peptides, thereby validating the reliability of virtual screening. Specifically, we conducted the docking of six peptides at the active site of CE to ascertain their optimal positions. Subsequently, we examined the interaction between these peptides and the critical amino acid residues of CE under these optimal docking conditions. Figure 4 furnishes an overview of the optimal conditions following the docking of cholesterol-lowering peptides and CE at the active site, and Table 2 provides a comprehensive account of the binding amino acids and the associated interaction forces.

During the study of the binding between CE and RK7, as shown in Figure 4a, RK7 established multiple interactions with amino acid residues Thr68, Ala67, Gln66, Asn118, Ala117, and Asn122 through hydrogen bonds. Notably, Thr68 formed two hydrogen bonds with RK7, with the bond to Ala117 being the shortest. The presence of this hydrogen bond not only enhanced the binding affinity of RK7 to CE, but also played a key role in RK7’s inhibitory effect, as shorter hydrogen bonds usually indicated tighter binding. Additionally, RK7 formed carbon–hydrogen bonds with amino acid residues Arg423, Pro425, and Asn122, and established alkyl, π–Alkyl, and π–π bonds through hydrophobic interactions with Met424, Leu69, and Phe119, while interacting with Ile426, Lys445, Tyr123, Leu124, Met424, Pro425, Tyr75, Leu69, Met111, and Gly112 via van der Waals forces. The diversity of these interactions indicated that RK7 bound to CE through various mechanisms, effectively inhibiting its activity. 

The molecular docking results for KQ7, as shown in Figure 4b, showed that KQ7 formed hydrogen bonds with CE amino acid residues His435, Asp437, Arg423, Met424, and Asn118, and a carbon–hydrogen bond with Ser422. Carbon–hydrogen bonds were generally weaker than hydrogen bonds but still contributed to binding strength. Furthermore, KQ7 formed π–Alkyl and alkyl bonds with Arg423, Met424, Ala117, and Phe119 through hydrophobic interactions, which played an important role in protein–ligand interactions, especially in less hydrophilic environments where such interactions significantly enhanced binding affinity [25]. It also interacted with Tyr427, Leu323, Gln440, Pro425, Leu120, Leu69, Gln71, Asn122, and Ala436 through van der Waals forces. These interactions suggested that KQ7’s binding mode in CE was complex and tight, aiding in the understanding of its inhibition mechanism.

In the study of QP13 Figure 4c, molecular docking results indicated that QP13 bound to CE amino acid residues Asp437 and Arg423 through hydrogen bonds, and to Ser422, Pro425, and Asn118 through carbon–hydrogen bonds. These carbon–hydrogen bonds may have stabilized the QP13–CE complex to some extent, complementing the hydrogen bonds. It formed π–Alkyl bonds with Ile426, Phe119, Ala117, Leu69, and Leu124 through hydrophobic interactions. QP13 also interacted with Asp434, Lys445, Asn118, Met424, Ile426, Tyr75, Thr68, Tyr427, Tyr123, Leu69, Gln440, Asn112, Phe119, and Arg63 through van der Waals forces, and formed electrostatic interactions with Asp72. QP13’s binding to important amino acid residues of CE, especially its hydrophobic residues (such as leucine, glycine, and proline), significantly contributed to the inhibition of CE, consistent with previous studies on bioactive peptides (Leu–Pro–Tyr–Pro) isolated from soybean glycinin, which demonstrated low cholesterol effects [26].

The molecular docking results for VN10, as shown in Figure 4d, showed that VN10 bound to CE with Arg423, Tyr75, and Gln71 through hydrogen bonds, and with Pro425 and Met424 through carbon–hydrogen bonds. VN10 also established electrostatic interactions with Met424 and Asp437 and formed π–π and π–Alkyl bonds with Ile426, Pro425, Met424, Arg423, Leu69, and Tyr75 through hydrophobic interactions. Π–Alkyl and Pi–Pi bonds enhanced VN10’s binding through hydrophobic effects, and these interactions may have helped stabilize the VN10–CE complex. VN10’s diverse interactions suggested its potential role in inhibiting CE. 

The docking results for TL9, as shown in Figure 4e, indicated that TL9 bound to CE through hydrogen bonds with Tyr75 and Arg423, a carbon–hydrogen bond with Ile426, and electrostatic interactions with Arg423. Simultaneously, TL9 formed π–Alkyl bonds with Leu69, Tyr75, Pro425, Met424, Ile426, and Ala117 through hydrophobic interactions, and interacted with Gln71, Thr68, Met111, Ala67, Arg63, Ser422, Asp437, Tyr427, Asn118, Phe119, and Thr68 via van der Waals forces. Although the van der Waals forces were relatively weak, their widespread presence helped to stabilize the overall protein–ligand complex structure. The binding characteristics of TL9 supported its potential as a CE inhibitor. 

The docking results for LQ10, as shown in Figure 4f, showed that LQ10 bound to CE with Arg423 and Tyr75 through hydrogen bonds, Pro425 and Tyr75 through carbon–hydrogen bonds, and formed π–Alkyl bonds with Leu69, Tyr75, and Ile426 through hydrophobic interactions. LQ10 also interacted with Tyr453, Pro425, Gly452, Met424, Thr68, Gln71, Ala117, and Tyr427 via van der Waals forces. The binding mode of LQ10 indicated that it had significant CE inhibitory activity. 

Docking studies of SN12, as shown in Figure 4g, indicated that it formed hydrogen bonds with CE residues Asn118 and Tyr75, a carbon–hydrogen bond with Asn118, and π–Alkyl and π–π bonds through hydrophobic interactions with Phe119, Ala117, Leu69, Met111, and Tyr75. SN12 also interacted with Leu274, Leu120, Ala113, Gly116, Gly112, Asp72, Thr74, Ala67, Tyr75, Arg63, Gln71, and Thr70 through van der Waals forces. The binding characteristics of SN12 further validated its inhibitory activity against CE. 

Atorvastatin, which is shown in Figure 4h, is a commonly used cholesterol-lowering drug, and it formed hydrogen bonds with CE amino acid residues Leu124 and Asn118, which may have helped stabilize the drug–enzyme complex and enhanced binding affinity [27]. Hydrogen bonds typically contributed to the stability of molecular structures, which was crucial for drug binding. The carbon–hydrogen bonds formed between atorvastatin and Asn118 may have enhanced the specificity of drug–enzyme binding, with this non-polar interaction contributing to increased binding stability. Van der Waals interactions between atorvastatin and various amino acid residues of CE, such as Lys445 and Phe119, were also crucial for Binding capacity of atorvastatin to CE. Van der Waals forces are short-range interactions, but they played an important role in stabilizing the drug–enzyme complex. Hydrophobic interactions, such as alkyl, π–Alkyl, and π–π bonds, generally involved the interaction of drug molecules with hydrophobic regions of the enzyme. Such interactions may have caused the drug to bind more tightly to the enzyme’s active site, thereby enhancing its inhibitory effect. Atorvastatin may have inhibited the activity of CE by interacting with relevant amino acid residues of the enzyme. CE played a crucial role in cholesterol metabolism, which was primarily responsible for the hydrolysis of cholesterol esters. The inhibitory effect of atorvastatin may have reduced cholesterol accumulation, thus, providing beneficial effects for lowering cholesterol levels. 

CE is an α/β-fold hydrolase, with its key active site containing a catalytic triad (serine–aspartate–histidine) and an oxyanion hole (glycine–alanine). These structural features were crucial for the inhibition activity of CE [28]. Cholesterol-lowering peptides bound to CE through various mechanisms, including hydrogen bonds, carbon–hydrogen bonds, hydrophobic interactions, and van der Waals forces. The formation of these interactions indicated that seven peptides bound to CE through multiple pathways, enhancing their binding capability. Therefore, peptides such as RK7, KQ7, QP13, and VN10 showed potential in inhibiting CE activity by binding to different amino acid residues of CE.

Liu et al. [29] found that hydrogen bonds are crucial for the stability of protein complexes. Hydrophobic, electrostatic, and van der Waals interactions promote peptide binding to proteins to reduce enzyme activity. RK7, KQ7, QP13, VN10, TL9, LQ10, and SN12 form 10, 7, 7, 5, 3, 5, and 3 hydrogen bonds with CE, respectively. KQ7 and QP13 each form two hydrogen bonds with Asn118. SN12 and atorvastatin each form one hydrogen bond and one carbon–hydrogen bond with Asn118. RK7 forms the shortest hydrogen bond with Ala117, resulting in tight binding to CE. TL9 forms a π–Alkyl bond with Ala117, LQ10 interacts with Ala117 via van der Waals forces, and SN12 forms a hydrophobic interaction with Ala117. Additionally, atorvastatin interacts with Ala117 through hydrophobic interactions. Ghosh [30] noted in early studies that His435 plays a crucial role in the inhibition of CE by KQ7 and VN10. They also found that acidic residues like Asp434 and Asp437 near His435 are essential for enhancing proton shuttle activity in CE’s catalytic function [31,32]. RK7, KQ7, QP13, VN10, and TL9 each form hydrogen bonds with Arg423 in CE. RK7 and atorvastatin form hydrogen and van der Waals bonds with Thr68, Ala67, Gln66, and Asn112. VN10 and atorvastatin each form hydrogen bonds with Tyr75 and Gln71. TL9, SN12, and atorvastatin all form hydrogen and van der Waals bonds with residues in CE. In addition to hydrogen bond interactions, KQ7 and VN10 also interact with Arg423 through hydrophobic interactions, while TL9 forms electrostatic interactions with Arg423. Although the peptides have minimal interaction with CE’s major hotspots, the study indicates that even standard inhibitors like DORSILURIN-F can bind to residues distant from these hotspots, such as Gly106, Gly112, Tyr125, Glu193, Trp227, Met281, Leu282, Val285, Ile323, Phe324, Ile327, and Leu329 [33].

Therefore, different drugs and peptides likely use various interaction patterns to inhibit CE. CE may lack specific binding sites for predicting inhibition. Thus, these peptides might act as allosteric inhibitors, modulating CE activity.

### 3.3. GROMACS Analysis of the Molecular Dynamics Stability of Peptide–CE Complexes

#### 3.3.1. RMSD, RMSF, Hydrogen Bonds, SASA, and Gibbs Free Energy Stability Analysis

The experiment evaluated the structural stability of protein molecules during the simulation period by analyzing the root-mean-square deviations (RMSD) in molecular dynamics simulations [34]. The RMSD value was a commonly used indicator to measure the stability of the protein structure. A low RMSD value usually indicated that the molecule maintained a relatively stable structure during the simulation process. As shown in Figure 5a, the experiment used the α-carbon atom RMSD of the complexes of CE and peptides (RK7, KQ7, QP13, VN10), and compared the structural changes of the complexes of peptides and CE during a 100 ns simulation process. The experimental results showed that the RMSD values of the complexes between each peptide and CE were relatively low, indicating that these peptides could form stable complexes with CE. During the initial 0–20 ns period of molecular dynamics simulation, the complex structure might have undergone a series of adjustments to adapt to thermodynamic conditions. These adjustments might have included the rearrangement of side chains and local reorganization of secondary structures. The significant fluctuations in the initial RMSD also reflected the adaptation process of the complex at the beginning of the simulation. The stability after this fluctuation indicated that the complex had reached a thermodynamically stable state after undergoing structural adjustments, which was also a common phenomenon in molecular dynamics simulations. The average RMSD values of peptides RK7, KQ7, QP13, and VN10 were 2.60 Å, 2.44 Å, 2.66 Å, and 2.34 Å, respectively, indicating that these peptides maintained good stability when bound to CE. Overall, the RMSD values fluctuated between 0.2 nm and 0.3 nm, indicating that peptides RK7, KQ7, QP13, and VN10 could form stable complex systems with CE. Studies have shown that the RMSD value of the complex was typically between 0.2 and 0.5 nm in a stable state, which was consistent with the results observed in this study [35]. Stable peptide–protein complexes were of great significance for disease prevention and intervention, as they might represent effective binding interfaces and potential inhibitors. 

The RMSF value reflected the amplitude of movement of amino acid residues. A high RMSF value indicated significant displacement of these amino acid residues during the simulation process, and the structure of these regions was usually more flexible. In contrast, low RMSF values indicated that these regions were relatively stable. As shown in Figure 5b, from the experimental results, it could be seen that when the four peptides (RK7, KQ7, QP13, VN10) bound to CE, specific regions (such as amino acid residues 265–285, 320–390, and 420–435) had higher RMSF values. This indicated that these regions had significant structural flexibility and might have played an important role in protein–ligand binding processes. Especially in the RK7 complex, the alpha helix and its adjacent circular regions exhibited very high fluctuations, suggesting that RK7 might have altered the functional state of CE by affecting the structural conformation of these regions. The binding of ligands to proteins often caused local or global structural changes in the protein. These changes might have led to an increase in the flexible region of the protein, thereby affecting the RMSF value [36]. During the binding process between CE and ligands (four peptides), the RMSF values in specific regions were higher, indicating greater flexibility and range of motion. A high RMSF value might have reflected structural rearrangement or dynamic changes in certain regions of the protein after ligand binding, which might have been active or regulatory sites of the protein. However, the average fluctuation range of RMSF values throughout the entire process was less than 0.2 nm, indicating that peptides RK7, KQ7, QP13, and VN10 maintained structural stability during the binding process with CE.

The research results from the laboratory were consistent with Santo’s study [35], which suggested that ligand binding typically led to increased local or global fluctuations in protein structure, which might be related to the affinity of ligand binding and the regulation of protein function.

As shown in Figure 5c, hydrogen bonds played a crucial role in maintaining the stability of protein–ligand complexes. They helped maintain the structure and function of proteins by providing additional stability. The changes in the hydrogen bond network in the complexes of CE with four peptides (RK7, KQ7, QP13, VN10) reflected the binding affinity and stability of different peptides to CE. The experimental results showed that the number of hydrogen bonds fluctuated between 2 and 12, indicating the dynamic changes of hydrogen bonds during ligand binding. This fluctuation might have been related to conformational changes of ligands during binding and adaptive adjustments of proteins. The fluctuation in the number of hydrogen bonds in the RK7–CE complex was relatively stable, indicating the formation of a strong and stable hydrogen bond network between RK7 and CE, thereby maintaining the stability of the complex.

The formation of hydrogen bonds typically involves interactions between hydrogen bond donors and acceptors, which could help stabilize the binding of proteins and ligands by enhancing interactions to reduce the system’s free energy. In molecular dynamics simulations, the dynamic changes of hydrogen bonds might have been related to the relative motion of proteins and ligands [36,37]. The binding of ligands not only directly changed the hydrogen bond network but might have also affected the stability of hydrogen bonds by inducing structural rearrangement or deformation of proteins. This rearrangement could have increased or decreased the number of hydrogen bonds, thereby affecting the overall stability of the complex [38]. By comparing the hydrogen bonds obtained during docking, a more comprehensive understanding of the impact of hydrogen bonds on the stability of protein–ligand interactions could be gained.

The solvent-accessible surface Area (SASA) represents the surface area of a protein that interacts with solvent molecules [39]. During the 100 ns molecular dynamics simulation, we calculated the SASA values for the interaction between CE and peptides RK7, KQ7, QP13, and VN10, and subsequently plotted these values (Figure 5d). The SASA trace exhibited a sharp increase within the first 20 ns, indicating structural relaxation. The average SASA values for RK7, KQ7, QP13, and VN10 were 228.01 nm^2^, 229.07 nm^2^, 230.41 nm^2^, and 225.63 nm^2^, respectively. The hydrophobic SASA showed considerable fluctuation within the 100 ns simulation, suggesting that the exposure of non-polar residues was influenced by the peptides. Peptides KQ7 and RK7 demonstrated higher SASA values than QP13 and VN10. Elevated SASA values reflected the structural flexibility of the enzyme in solution. The greater the extent of solvent contact with non-polar residues of CE in complexes with RK7, KQ7, QP13, and VN10, the higher the likelihood of interaction with these peptides.

Figure 5(e-1,e-2) presents the 3D and 2D Gibbs energy landscape of the CE–RK7 complex, respectively. The complex exhibited low Gibbs free energy when the radius of gyration (Rg) value was 2.30–2.39 nm and the RMSD value was 0.11–0.28. Figure 5(f-1,f-2) presents the 3D and 2D Gibbs energy landscape of the CE–KQ7 complex, respectively. The complex exhibited low Gibbs free energy when the Rg value was 2.30–2.37 nm and the RMSD value was 0.11–0.28. Figure 5(g-1,g-2) presents the 3D and 2D Gibbs energy landscape of the CE–QP13 complex, respectively. The complex exhibited low Gibbs free energy when the Rg value was 2.30–2.38 nm and the RMSD value was 0.21–0.26. Figure 5(h-1,h-2) presents the 3D and 2D Gibbs energy landscape of the CE–VN10 complex, respectively. The complex exhibited low Gibbs free energy when the Rg value was 2.30–2.37 nm and the RMSD value was 0.09–0.14.

The low Gibbs free energy of a complex is usually related to its stability in a specific conformation. These low-energy states may be due to effective interactions between proteins and ligands (such as hydrogen bonding, hydrophobic interactions, and electrostatic interactions) as well as the rational folding and compact structure of proteins [40].

#### 3.3.2. MM/GBSA Calculations and Analyses

MM/GBSA is a method for estimating the free energy in silico for studies of ligands in protein complexes [41]. In this study, we used the electrostatic energy obtained from molecular mechanics force field (MM) calculations and generalized Born (GB) calculations of the electrostatic contribution to the solvation free energy [42]. The binding free energies between the peptides RK7, KQ7, QP13, VN10, and CE were calculated without considering entropic contributions.

Table 3 shows that the ΔGMMGBSA values of the peptide complexes containing RK7, KQ7, QP13, and VN10 are −61.34 kcal/mol ± 4.02, −65.24 kcal/mol ± 4.69, −39.97 kcal/mol ± 10.66, and −1.09 kcal/mol ± 7.52, respectively, with roughly consistent contributions from van der Waals (VDW) interactions. The binding free energies of the RK7 (−61.34 kcal/mol ± 4.02) and KQ7 (−65.24 kcal/mol ± 4.69) complexes are lower than those of the QP13 (−39.97 kcal/mol ± 10.66) and VN10 (−1.09 kcal/mol ± 7.52) complexes, indicating that the RK7 and KQ7 complex systems have lower energy and are, hence, more stable.

#### 3.3.3. Analysis of Interactions in the 0–100 ns Process of Molecular Dynamics

As shown in Figure 6, RK7, KQ7, QP13, and VN10 formed stable complexes with CE at 0 ns through hydrophobic interactions and hydrogen bonding. These interactions were essential for the stability of the complex. Even in molecular dynamics simulations up to 100 ns, there was no significant change in the overall structure, indicating that these protein complexes were able to maintain their structures in a dynamic environment. The simulations suggested that although the overall structure of the complexes did not change significantly over 100 ns, the increase in salt bridge interactions might have indicated that the protein complexes underwent adaptive changes during the simulations to enhance their stability or functional affinity. This might have affected the function of the protein or its behavior within the cell. It had been suggested that hydrogen bonding and salt bridge interactions in protein complexes might undergo significant changes during molecular dynamics simulations, which was consistent with experimental observations [43,44,45]. These changes were usually associated with functional regulation or changes in the stability of the proteins. In summary, this experiment analyzed the interactions of RK7, KQ7, QP13, VN10, and CE between 0 and 100 ns using molecular docking and molecular dynamics simulations. The results showed that the protein complexes remained stable during the 100 ns simulation and that the increase in salt bridge interactions might have indicated adaptive changes and greater stability of the protein complexes during the simulation. These results validated the predictions of molecular docking and molecular dynamics simulations and provided new insights into understanding the stability of protein-peptide interactions and the biological mechanisms that might have been involved.

### 3.4. Synthesis and Validation of CE Inhibitory Activity of Peptides RK7, KQ7, QP13 and VN10

#### 3.4.1. Synthesis of Peptides RK7, KQ7, QP13 and VN10

The theoretical relative and experimentally measured molecular masses of RK7 were 875.06 Da and 874.90 Da, respectively. The theoretical relative and experimentally measured molecular masses of KQ7 were 779.96 Da and 779.50 Da, respectively. Both molecules showed molecular masses close to the theoretical values, with purities of 98.07% for RK7 and 99.76% for KQ7. The theoretical relative and experimentally measured molecular masses of QP13 were 1392.61 Da and 1392.60 Da, respectively. The theoretical relative and experimentally measured molecular masses of VN10 were 1100.27 Da and 1100.30 Da, respectively. Both QP13 and VN10 exhibited molecular masses close to the theoretical values, with purities of 98.10% for QP13 and 98.20% for VN10. The synthesis results of all peptides met the expected standards. The synthesized peptides RK7, KQ7, QP13, and VN10 were used in the experiments for the verification of CE inhibition activity.

#### 3.4.2. Validation of In Vitro Activity of Peptides RK7, KQ7, QP13, and VN10

In this experiment, we used UV spectrophotometry to determine the inhibitory activity of peptides RK7, KQ7, QP13, and VN10 on CE. The experimental results are shown in Figure 7. The inhibition rates of peptides RK7, KQ7, QP13, and VN10 on CE increased with the increase of peptide mass concentration. The inhibition curve of atorvastatin on CE was y = −28.259x^2^ + 81.97x + 15.422, R^2^ = 0.9683, with inhibition rates ranging from 28.78% to 58.63% at concentrations of 0.2–1.0 mg/mL. At a concentration of 0.2–1.0 mg/mL, the inhibition rate was 63%. The curve of the peptide RK7’s inhibition on cholesterol esterase was y = 0.9142x^2^ + 49.52x + 11.618, R^2^ = 0.9932, with its inhibition ranging from 22.24% to 61.16% at concentrations of 0.2–1.0 mg/mL. The curve of the peptide KQ7’s inhibition on cholesterol esterase was y = −27.875x^2^ + 74.466x + 13.428, R^2^ = 0.9967, with its inhibition ranging from 28.78% to 58.63% at concentrations of 0.2–1.0 mg/mL. The inhibition curve of peptide QP13 on CE was y = −5.9865x^2^ + 54.753x + 14.556, R^2^ = 0.9910, with inhibition rates ranging from 25.86% to 62.37% at concentrations of 0.2–1.0 mg/mL. The inhibition curve of peptide VN10 on CE was y = −12.386x^2^ + 60.451x + 10.095, R^2^ = 0.9957, with inhibition rates ranging from 21.32% to 58.09% at concentrations of 0.2–1.0 mg/mL.

The CE Inhibition rates of peptides RK7, KQ7, QP13, and VN10 were significantly higher than those of the control group at all concentrations (*p* < 0.05). According to the inhibition curve, the IC_50_ values of RK7, KQ7, QP13, and VN10 were 8.16 × 10^−7^ mol/L, 8.10 × 10^−7^ mol/L, 4.63 × 10^−7^ mol/L, and 7.97 × 10^−7^ mol/L, respectively, while atorvastatin was 8.54 × 10^−7^ mol/L. In terms of the inhibitory activity of CE, further in vitro validation experiments showed that RK7, KQ7, QP13, and VN10 all had good inhibitory effects on CE, and their inhibitory effects on CE increased with the increase of peptide mass concentration. The MM/GBSA value of KQ7 was −65.24 Kcal/mol ± 4.69, and the IC_50_ value was 8.10 × 10^−7^ mol/L. Therefore, by integrating molecular dynamics and in vitro validation experiments, it could be inferred that KQ7 had the highest inhibitory activity on CE. In vitro activity validation experiments further demonstrated that peptides RK7, KQ7, QP13, and VN10 all had good inhibitory effects on CE, and the inhibitory effect of CE increased with the increase of peptide mass concentration. Different amino acid compositions and peptide sequences affected the biological activity of peptides, and hydrophobic amino acids had a significant impact on CE inhibitory activity. Studies had shown that the inhibitory effect on CE was mainly caused by hydrogen bonding and hydrophobic interactions at its active sites [46,47].

Compared to existing statins, peptides RK7, KQ7, QP13, and VN10 also demonstrated outstanding performance in molecular docking and kinetic simulations. Statins such as atorvastatin exhibited high inhibitory activity by inhibiting CE through multiple interactions. Compared to RK7, KQ7 exhibited stronger potential inhibitory activity due to its high hydrophobicity and more rapidly stable binding state. Additionally, the results from in vitro experiments indicated that KQ7 exhibited higher inhibitory activity against CE compared to RK7, which might have been related to the more stable hydrogen bonding network formed during the binding process. Peptides RK7, KQ7, QP13, and VN10 interfered with the normal function of enzymes and prevented cholesterol synthesis by tightly binding to CE. This inhibitory effect mainly relied on hydrophobic interactions and hydrogen bonding between peptides and enzymes. In addition, electrostatic interactions and van der Waals forces also played an important role in stabilizing the composite. These findings provided new ideas for the future development of peptide-based CE inhibitors.

#### 3.4.3. Simulating the Effect of Gastrointestinal Digestion on CE Inhibition Activity

Figure 8 shows the changes in the CE inhibition rates of peptides RK7, KQ7, QP13, and VN10 after simulated gastrointestinal digestion. As shown in the figure, the CE inhibition rates of RK7, KQ7, QP13, and VN10 increased with time after digestion by pepsin and trypsin. Specifically, within 120 min of simulated gastrointestinal digestion, the CE inhibition rate of RK7 ranged from 43.00% to 61.56%, with significant changes observed (*p* < 0.05). This indicated that RK7 maintained a certain level of CE inhibition activity during digestion, suggesting good biological stability under pepsin and trypsin degradation. The ability of RK7 to retain its CE inhibition activity during digestion might have been due to its relatively stable structure or the retention of some inhibitory activity in its degradation products. RK7’s backbone might have had strong resistance to pepsin and trypsin, or active fragments in its degradation products might have still effectively bound to CE and maintained inhibition. This stability was significant for RK7 as a potential therapeutic agent or supplement against cholesterol esterase; KQ7 showed a significant increase in CE inhibition activity at 60 min during digestion (*p* < 0.05), with a CE inhibition rate reaching 58.63% at 120 min. The significant increase in KQ7’s inhibition activity during digestion might have been due to the cleavage of KQ7 into peptide fragments with higher inhibitory activity under the action of digestive enzymes. This cleavage might have exposed more active sites, enhancing KQ7’s binding and inhibition ability against CE. This suggested that KQ7 might have experienced increased biological activity during digestion, which could have helped lower cholesterol levels; both QP13 and VN10 also showed significant increases in CE inhibition activity during digestion. Particularly at 60 min, both peptides exhibited significant increases in inhibition activity (*p* < 0.05), indicating that the inhibition activity of peptides QP13 and VN10 increased with digestion, possibly due to the release of active peptide fragments caused by pepsin and trypsin. These fragments might have had stronger biological activity, thereby enhancing the inhibition effect on CE. Their inhibition rates ranged from 40.59% to 58.10%, indicating that these peptides could have maintained or enhanced their biological activity in the digestive environment, potentially affecting cholesterol metabolism. These experimental results suggested that RK7, KQ7, QP13, and VN10 could effectively inhibit CE after gastrointestinal digestion, with significant enhancement of inhibition activity observed particularly for KQ7 and RK7. These peptides exhibited good stability and enhanced inhibition activity in the digestive system, providing a foundation for their potential application as anti-cholesterol treatments or supplements. Future research could further explore the mechanisms of activity changes for these peptides in different digestive environments.

## 4. Conclusions

CE was an important enzyme involved in various biological processes, including protein digestion and degradation. By inhibiting the activity of CE, these biological processes could be regulated, which might have impacted the treatment of high cholesterol diseases. The experimental results indicated that the effective inhibitory effects of peptides RK7, KQ7, QP13, and VN10 suggested that these peptides might have potential medicinal value. The molecular docking results showed that the binding of peptides RK7, KQ7, QP13, and VN10 to CE stabilized the structure of the complexes through interactions such as hydrogen bonding. This meant that the peptides inhibited the enzymatic activity of CE by forming strong bonds with specific amino acid residues such as His435, Asn118, etc. This inhibitory effect was related to the structure and amino acid composition of the peptides, which interfered with the substrate binding of CE through competitive mechanisms. The binding stability between peptides and CE was further validated by molecular dynamics simulation analysis of parameters such as RMSD (root-mean-square deviation), RMSF (root-mean-square fluctuation), hydrogen bonding, SASA (solvent-accessible surface area), Gibbs free energy, and MM/GBSA (molecular mechanics/generalized Born surface area). Peptides RK7, KQ7, QP13, and VN10 showed significant inhibitory effects on CE in vitro experiments, with IC_50_ values of 8.16 × 10^−7^ mol/L, 8.10 × 10^−7^ mol/L, 4.63 × 10^−7^ mol/L, and 7.97 × 10^−7^ mol/L, respectively. Based on the comprehensive experimental results and analysis, it was concluded that among the four peptides, peptide KQ7 exhibited the highest inhibitory activity against CE. Atorvastatin showed the strongest inhibitory effect on CE in the inhibition experiment, with an IC_50_ of 8.54 × 10^−7^ mol/L, indicating its strongest inhibitory ability. As a control drug, atorvastatin’s significant inhibitory effect validated the reliability of the experimental method and provided a benchmark for the inhibitory effects of other peptides. After 120 min of gastrointestinal digestion, peptides RK7, KQ7, QP13, and VN10 were all able to effectively inhibit CE, with inhibition rates ranging from 40.59% to 62.37%. Notably, the inhibition activities of peptides KQ7 and RK7 against CE were significantly enhanced after simulated gastrointestinal digestion. Due to the effective inhibition of CE by peptides RK7, KQ7, QP13, and VN10, it was suggested that they might serve as functional ingredients in the field of biomedicine for the prevention of related diseases or as a basis for studying natural foodborne inhibitors. These experimental results provided an important theoretical basis for further research, which could help develop new CE inhibitors, improve existing treatment strategies, and explore the possible roles of peptides in other biological processes. This study validated the inhibitory mechanism of peptides RK7, KQ7, QP13, and VN10 on CE through bioinformatics tools and in vitro experiments, indicating that these peptides could exert their effects by stably binding and inhibiting CE activity. The control results of atorvastatin further validated the effectiveness of the experiment and provided important references for the development and biological research of future foodborne natural inhibitors.

## Figures and Tables

**Figure 1 foods-13-02970-f001:**
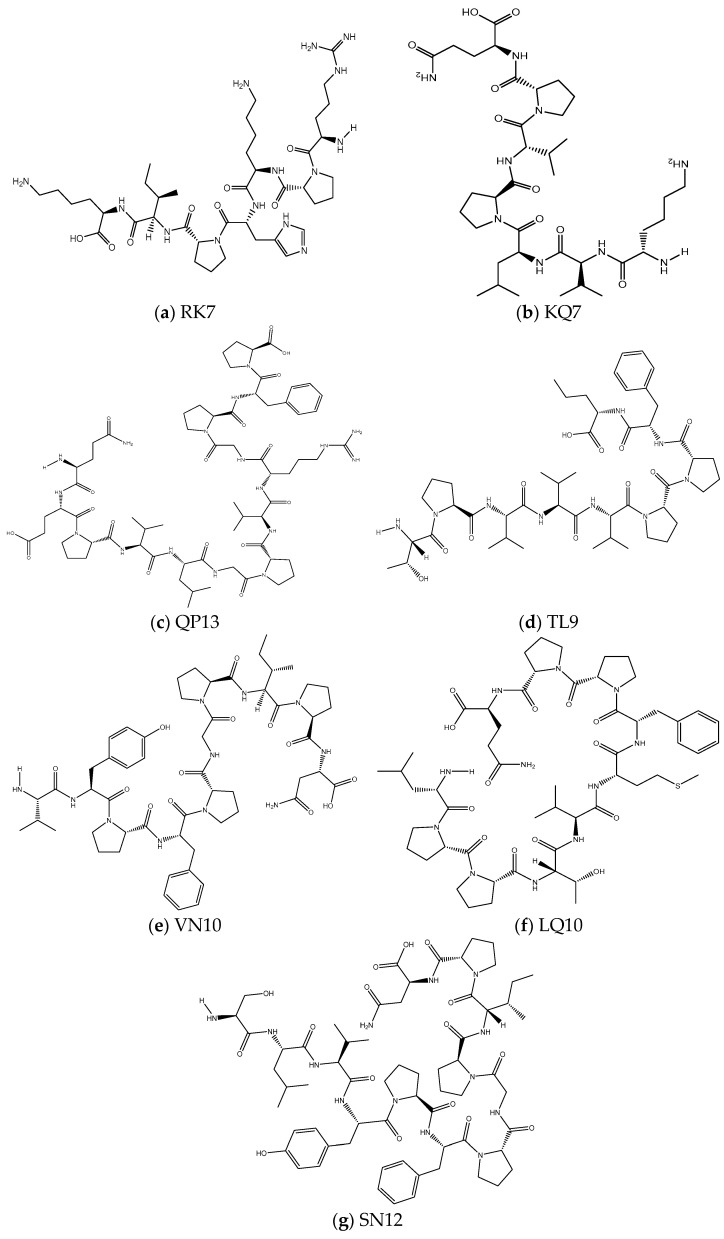
The two-dimensional molecular structures of (**a**) RK7, (**b**) KQ7, (**c**) QP13, (**d**) TL9, (**e**) VN10, (**f**) LQ10, (**g**) SN12.

**Figure 2 foods-13-02970-f002:**
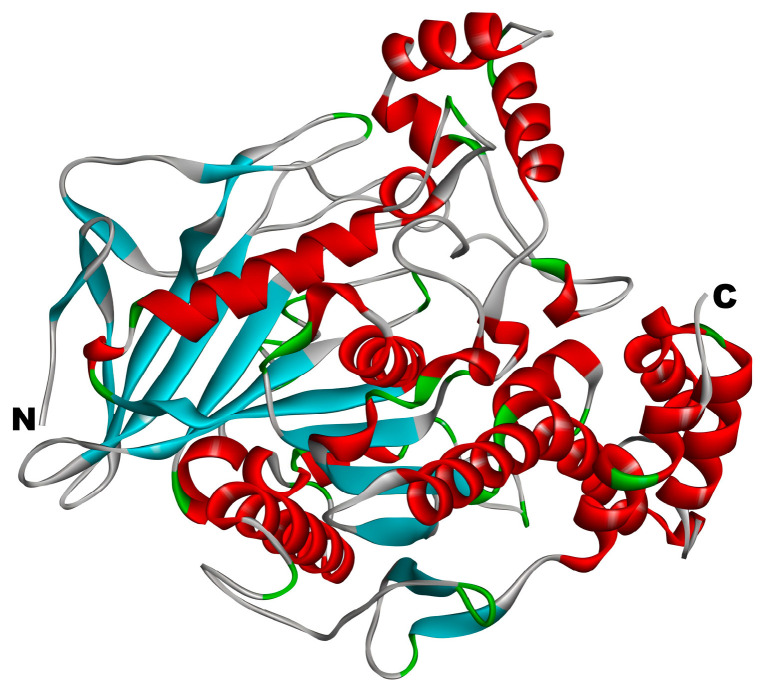
Three-dimensional structure of cholesterol esterase reductase (PDB:1AQL); the N-terminal and C-terminal positions in the structural domain of cholesterol esterase are la-belled in the figure.

**Figure 3 foods-13-02970-f003:**
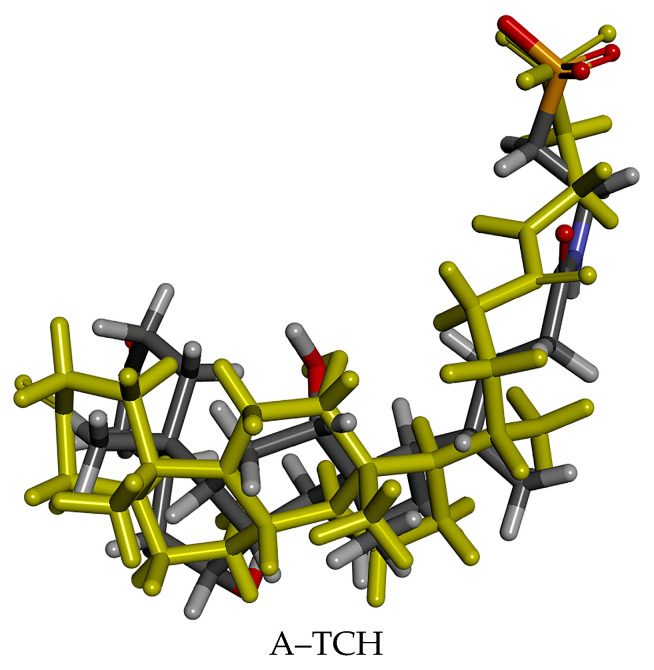
Superimposed comparison of ligand conformation in the original crystals with that of the docked ligand in the target protein 1AQL co-crystal complex.

**Figure 4 foods-13-02970-f004:**
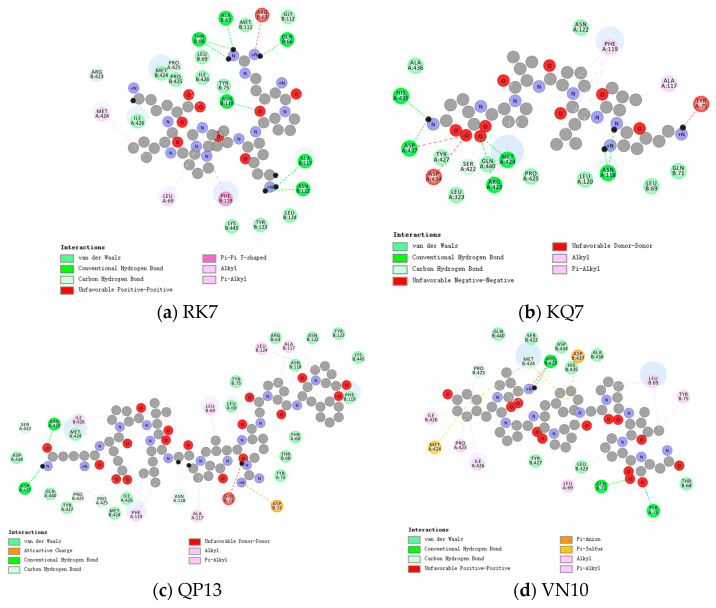
Molecular docking of peptides (**a**) RK7, (**b**) KQ7, (**c**) QP13, (**d**) VN10, (**e**) TL9, (**f**) LQ10, (**g**) SN12, and (**h**) atorvastatin with CE: a close view of the active site binding with peptides (right). Key residues that interacted with peptides are shown in stick and colored green. Green dotted lines represent hydrogen bonding. The red dotted line is the Unfavourable Donor-Donor Interaction. Pink dashed lines are hydrophobic effects. For interpretation of the references to color in this figure legend, the reader is referred to the color coding in the figure).

**Figure 5 foods-13-02970-f005:**
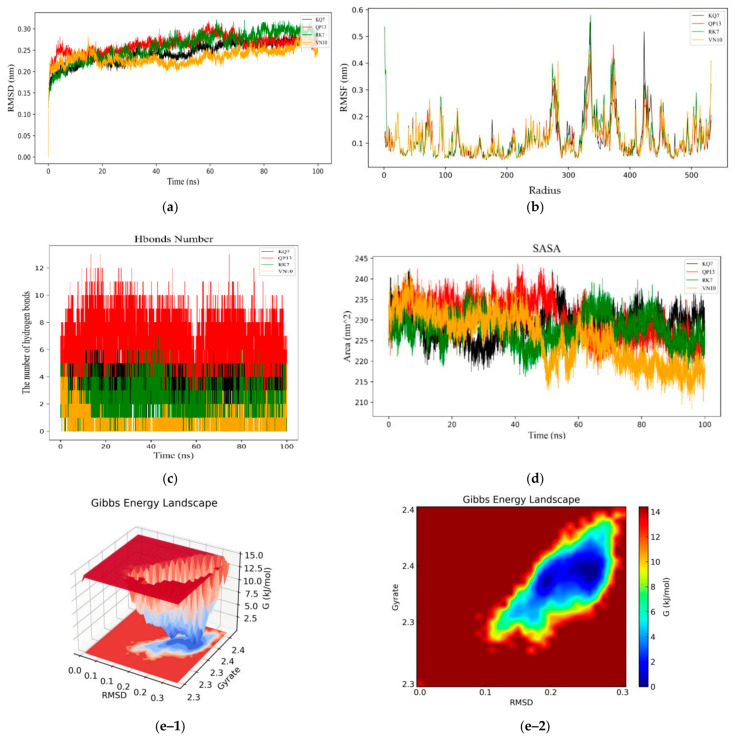
Molecular dynamics simulation analysis of the CE–peptides complex. (**a**) RMSD curve of KQ7 (black line), RK7 (green line), QP13 (red line), and VN10 (yellow line). (**b**) RMSF curve of KQ7 (black line), RK7 (green line), QP13 (red line), and VN10 (yellow line). (**c**) Hydrogen bonds curve of ACHE. (**d**) SASA of KQ7 (black line), RK7 (green line), QP13 (red line), and VN10 (yellow line). (**e-1**,**e-2**;**f-1**,**f-2**;**g-1**,**g-2**;**h-1**,**h-2**) Three-dimensional and two-dimensional Gibbs free energy landscape of the CE–Peptides (RK7, KQ7, QP13, VN10) complex.

**Figure 6 foods-13-02970-f006:**
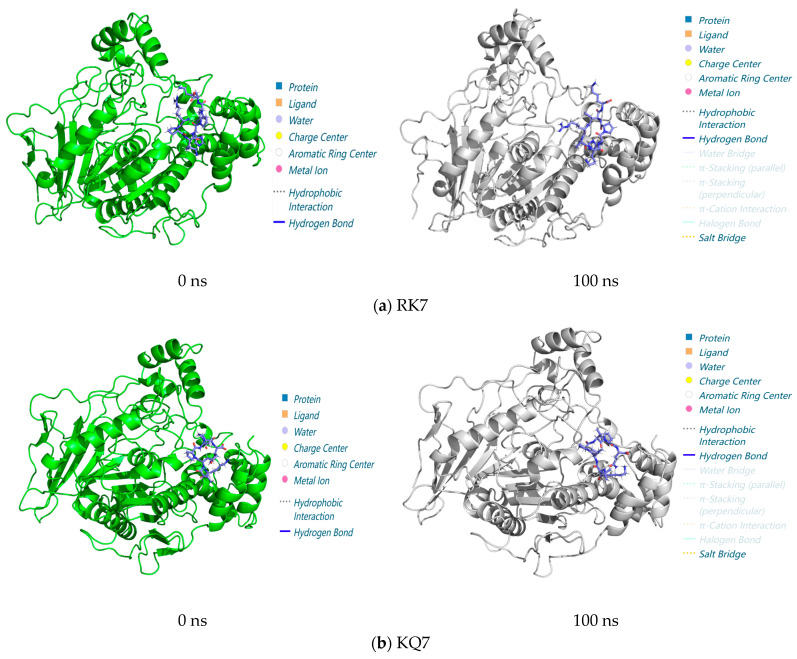
Structure of peptide interaction with CE in the 100 ns molecular dynamics regime: The 3D interaction maps of (**a**) RK7, (**b**) KQ7, (**c**) QP13, and (**d**) VN10 obtained at 0 ns and 100 ns, with elaboration on the interaction diagrams and associated forces between the protein and peptide at 0 ns and 100 ns.

**Figure 7 foods-13-02970-f007:**
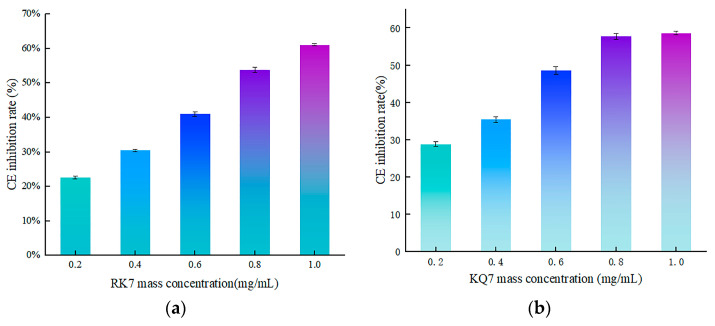
The effect of peptide mass concentration and atorvastatin on CE inhibition rate. (**a**) Effect of RK7 mass concentration on CE inhibition rate. (**b**) Effect of KQ7 mass concentration on CE inhibition rate. (**c**) Effect of QP13 mass concentration on CE inhibition rate. (**d**) Effect of VN10 mass concentration on CE inhibition rate. (**e**) Effect of Atorvastatin mass concentration on CE inhibition rate.

**Figure 8 foods-13-02970-f008:**
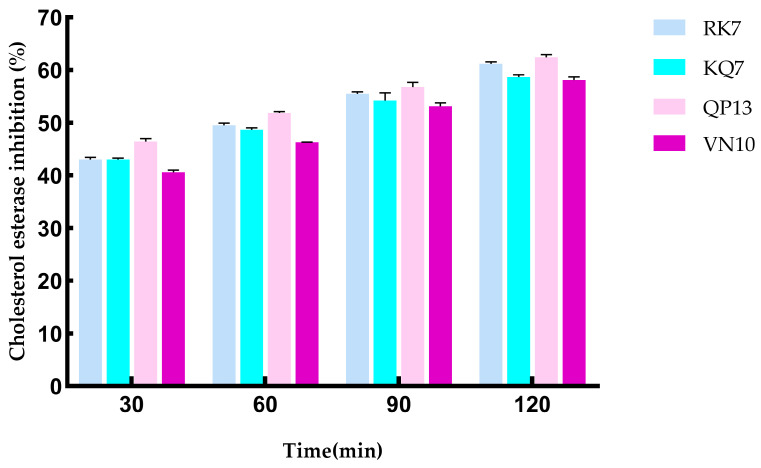
Effect on CE inhibitory activity after simulated gastrointestinal digestion.

**Table 1 foods-13-02970-t001:** Physicochemical properties of peptides.

Peptide Sequence	Molecular Weight/(Da)	Isoelectric Point	Net Charge	Hydrophobic Amino Acids	Proportion of Hydrophobic Amino Acids
RK7	874.90	11.57	3	P, I	42.86%
KQ7	779.50	9.84	1	V, L, P	71.42%
QP13	1392.60	6.58	0	P, V, L, G, F	61.54%
TL9	968.19	3.32	0	P, V, F, L	88.89%
VN10	1100.26	3.63	0	V, P, F, G, I	70.00%
LQ10	1300.50	4.34	0	L, P, V, M, F	66.67%
SN12	1126.37	3.70	0	L, V, P, F, G, I	80.00%

**Table 2 foods-13-02970-t002:** Binding amino acids and associated interaction forces of cholesterol-lowering peptides and CE after docking at the active site.

	RK7	KQ7	QP13	VN10	TL9	LQ10	SN12
Hydrogen bonds	Thr68 Ala67 Gln66 Asn118 Ala117 Asn122 Arg423 Pro425 Asn122	His435 Asp437 Arg423 Met424 Asn118 Ser422	Asp437 Arg423 Ser422 Pro425 Asn118	Arg423 Try75 Gln71 Pro425 Met424	Try75 Arg423 Ile426	Arg423 Tyr75 Pro425 Tyr75	Asn118 Tyr75 Asn118
Hydrophobic interactions	Met424 Leu69 Phe119	Arg423 Met424 Ala117 Phe119	Ile426 Phe119Ala117 Leu69 Leu124 Ala117	Ile426 Pro425Met424 Arg423 Leu69 Tyr75	Leu69 Tyr75 Pro425Met424 Ile426 Pro425 Ile426 Ala117	Leu69 Tyr75 Ile426	Phe119 Ala117 Leu69Met111 Tyr75
Van der Waals forces	Ile426 Lys445 Tyr123 Leu124 Met424 Pro425 Tyr75 Leu69 Met111 Gly112	Tyr427 Leu323 Gln440 Pro425 Leu120 Leu69 Gln71 Asn122 Ala436	Asp434 Lys445 Asn118 Met424 Ile426 Tyr75 Thr68 Tyr427 Tyr123 Leu69 Gln440 Asn112 Phe119 Arg63	Gln440 Ser422 Asp434 His435 Ala436 Thr68 Leu323 Tyr427	Gln71 Thr68 Met111 Ala67 Arg63 Ser422 Asp437 Tyr427 Asn118 Phe119 Thr68	Tyr453 Pro425 Gly452 Met424 Thr68 Gln71 Ala117 Tyr427	Leu274 Leu120 Ala113 Gly116 Gly112 Asp72 Thr74 Ala67 Tyr75 Arg63 Gln71 Thr70
Electrostatic interactions	—	—	Asp72	Met424Asp437	Arg423	—	—

Comment: The “—” symbol denotes the absence of amino acid residues involved in this interaction force.

**Table 3 foods-13-02970-t003:** MM/GBSA results of CE with RK7, KQ7, QP13 and VN10.

(kcal/mol)	RK7	KQ7	QP13	VN10
VDWAALS	−89.74 ± 0.01	−74.74 ± 1.38	−84.13 ± 2.79	−68.20 ± 0.15
ΔEEL	−25.73 ± 3.86	−72.67 ± 3.27	−50.58 ± 7.24	−152.73 ± 1.87
ΔEGB	65.11 ± 1.11	91.60 ± 3.06	106.08 ± 7.32	229.91 ± 7.28
ΔEsurf	−10.99 ± 0.01	−9.44 ± 0.20	−11.34 ± 0.02	−10.06 ± 0.11
ΔGgas	−115.46 ± 3.86	−147.40 ± 3.55	−134.71 ± 7.75	−220.94 ± 1.87
ΔGsolv	54.12 ± 1.11	82.16 ± 3.07	94.74 ± 7.32	219.84 ± 7.28
ΔTotal	−61.34 ± 4.02	−65.24 ± 4.69	−39.97 ± 10.66	−1.09 ± 7.52

Notes: ΔVDWAALS: van der Waals energy; ΔEEL: electrostatic energy; ΔEGB: polar solvation energy; ΔEsurf: non-polar solvation energy; ΔGgas: energy of the molecular mechanics term (gas-phase energy); ΔGsolv: solvation energy; ΔTotal: combined total energy.

## Data Availability

The original contributions presented in the study are included in the article, further inquiries can be directed to the corresponding author.

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
