# Peer review of "Study on the Inhibitory Effect of Bioactive Peptides Derived from Yak Milk Cheese on Cholesterol Esterase"

_foods, 2024, doi:10.3390/foods13182970_

Round 1

Reviewer 1 Report

Comments and Suggestions for Authors

The paper wrote about studying the bioactive peptides derived from yak milk that exhibited cholesterol-lowering properties. They searched the bioactive peptides derived from yak milk that exhibited cholesterol-lowering properties. They used molecular docking and dynamics technologies and the four peptides were synthesized to verify their 20 CE-inhibitory effects in vitro. Basically, the paper is interested in broad readers for yak milk’s diet. However, some data had to be improved for clear scientific meaning. First, they changed to concentration expression mg/ml to mol/L or M concentration. And some figures have to be redrawn for clear understanding. I summarized them below.

Main points

Please confirm that you can show the activity using real protein CE. (You should at least test the effectiveness of the best peptides with real protein CE.)

You had better redraw all figures for clear understanding. And please the result is more concise (Specially Figure 4  Figure 6 etc). Please suggest to use supplementary figures.

Figure 1 please display the primary sequence and 1st amino acids (schematic diagrams) and more information on figure legends.

Figure 2 Please rotate the degree and show the N-terminal (C-terminal). Describe more information about figure legends (ex: source human or PDB ID)

Figure 4, 6 The figure size is small and figure’s meaning is not clear.

Comments on the Quality of English Language

Moderate editing of English language is required. 

Reviewer 2 Report

Comments and Suggestions for Authors

Study on the inhibitory effect of bioactive peptides derived 2 from yak milk on cholesterol esterase

I recommend that the authors pay special attention to the in silico analysis and molecular docking. This would improve the discussion and conclusions.

The word in vitro, in silico, could be in italics because it is a Latin expression.

Line 11. The word peptide is repeated.

32 The acronym CVD has already been cited earlier on line 29

48 Shouldn't it be bioactive peptides instead of active peptides?

55 The acronym ACE hasn't been described before

Line 80. Generally, in addition to the brand name of the equipment, the model and country of origin are included.

Line 109. In Figure 1, the identification names for the peptides are missing: a) RK7 and b) KQ7

Line 111. In the in silico analysis of the 1AQL protein, the enzyme needs to be energetically optimized to further refine the structure.

119 Figure. 2 must be Figure 2.

Line 120. In the molecular docking it is not mentioned if it was specific, by batches or “blind”.

121 Is the Discovery Studio docking software the same as Discovery Studio Client v16.1.0 (DS)? It confuses me because they have different names

141 Na+  (symbol + as superscript)

146 The units (ps) and (fs) could be described as was done with nanoseconds

147 wich software was used?

Equation (1) must be receptor instead or reseptor

Table 1. Capital letter in the first column

Line 193. They mention that the peptides studied have a good amount of hydrophobic amino acids, to complete the information it is necessary to mention the percentage of the same in each peptide or to mention that the percentages are observed in Table 1.

205 capital letters after the dot

Figure.

Reviewer 3 Report

Comments and Suggestions for Authors

Several issues are noted in this mansucript by Peng Wang and co-authors, as detailed below:

1. It is unclear if the peptides used in this study were actually derived from yak milk (if so, how were these  isolated?) or synthetic analogues of yak milk peptides. This is a critical matter, and must be c;early stated in both the Abstract and the Introduction.

2. In Materials and Methods, it is mentioned that 4 of these peptides (RK7, KQ7, VN10, and QP13) were obtained from a particular company. Where were the other 3 peptides obtained from? 

3. The biggest issue with this study is the lack of consideration given to potential effects of gastro-intestinal digestion upon these peptides. Many of these peptides are longer (10-13 amino acids) than typical bioactive peptides (2-7 amino acids) used in both in vitro and in vivo experiments. It is likely that the larger peptides would be broken down into smaller fragments by gastro-intestinal digestion, and it remains to be determined if their effects on CE would be maintained. It is recommended that the authors perform simulated gastro-inetstinal digestion in vitro on these peptides, and study similar biological actions in the post-digestion fragments (i.e., on smaller peptides).

4. Finally, the effects on CE appear at relatively high concentrations, is it plausible to each such levels in the human body following consumption of yak milk/cheese? This should be disucssed in the Discussion section

Comments on the Quality of English Language

It is acceptable.
